# Bypassing Skip-Gram Negative Sampling: Dimension Regularization as a More Efficient Alternative for Graph Embeddings

## Abstract

A wide range of graph embedding objectives decompose into two components: one that attracts the embeddings of nodes that are perceived as similar, and another that repels embeddings of nodes that are perceived as dissimilar. Without repulsion, the embeddings would collapse into trivial solutions. Skip-Gram Negative Sampling (SGNS) is a popular and efficient repulsion approach that prevents collapse by repelling each node from a sample of dissimilar nodes. In this work, we show that when repulsion is most needed and the embeddings approach collapse, SGNS node-wise repulsion is, in the aggregate, an approximate re-centering of the node embedding dimensions. Such dimension operations are much more scalable than node operations and yield a simpler geometric interpretation of the repulsion. Our result extends findings from self-supervised learning to the skip-gram model, establishing a connection between skip-gram node contrast and dimension regularization. We use this observation to propose a flexible algorithm augmentation framework that improves the scalability of any existing algorithm using SGNS. The framework prioritizes node attraction and replaces SGNS with dimension regularization. We instantiate this generic framework for LINE and node2vec and show that the augmented algorithms preserve downstream link-prediction performance while reducing GPU memory usage by up to 33.3% and training time by 22.1%. Further, for graphs that are globally sparse but locally dense, we show that removing repulsion altogether can improve performance, but, when repulsion is otherwise needed, dimension regularization provides an effective and efficient alternative to SGNS.

## 1 Introduction

Graph embedding algorithms use the structure of graphs to learn node-level embeddings. Across unsupervised and supervised graph embedding algorithms, their loss functions serve the two roles of preserving *similarity* and *dissimilarity*. Nodes that are similar in the input graph should have similar embeddings, while dissimilar nodes should have dissimilar embeddings (Böhm et al., 2022). The push and pull of the similarity and dissimilarity objectives are key: in the absence of a dissimilarity objective, the loss would be minimized by embedding all nodes at a single embedding point, a degenerate and useless embedding. Often, enforcing dissimilarity is much more expensive than similarity, owing to the generally sparse nature of graphs and the number of pairs of dissimilar nodes growing quadratically with the size of the graph. Enforcing dissimilarity is also complex for graphs because graph data frequently have missing edges or noise (Young et al., 2021; Newman, 2018). In this paper, we show that while many past works have focused on repelling pairs of dissimilar nodes, the repulsion can be replaced with a regularization of the embedding dimensions, which is more scalable.

The skip-gram (SG) model is one of the most popular approaches to graph embeddings (Ahmed et al., 2013; Yang et al., 2024) and can be decomposed into preserving similarity and dissimilarity. Further, skip-gram negative sampling (SGNS) is a dominant method to efficiently approximate dissimilarity preservation. Instead of repelling all pairs of dissimilar nodes, SGNS repels only a sample of dissimilar nodes per pair of similar nodes. SGNS is utilized in LINE Tang et al. (2015) and node2vec Grover & Leskovec (2016), for instance, and has been shown to yield strong down-

stream performance. However, several analytical issues with SGNS have also been identified. First, SGNS introduces a bias by re-scaling the relative importance of preserving similarity and dissimilarity (Rudolph et al., 2016). Second, with SGNS, in the limit as the number of nodes in the graph approaches infinity, the similarities among embeddings diverge from the similarities among nodes in the graph (Davison & Austern, 2023). Although SGNS has been used to learn both graph and word embeddings (Mikolov et al., 2013; Mimno & Thompson, 2017), we focus on the graph context because, for graph embeddings in particular, SGNS remains a popular method for preserving dissimilarity (Chami et al., 2022).

In this paper, we propose a change in perspective and show that node repulsion in the SG model can be achieved via dimension centering. If $X$ is an embedding matrix where the rows are node embeddings, "dimensions" refers to the columns of $X$. We draw inspiration from recent advances in the self-supervised learning (SSL) literature, which show an equivalence between sample-contrastive learning and dimension-contrastive learning (Garrido et al., 2022; Bardes et al., 2022). Sample-contrastive learning explicitly repels dissimilar pairs while dimension-contrastive learning repels the dimensions from each other.

The known parallels between sample and dimension contrast, however, do not suggest whether SG loss functions can be also re-interpreted from the dimension perspective. In this paper, we begin by characterizing the degenerate embedding behavior when the dissimilarity term is removed altogether. We prove that, under mild initialization conditions, when only positive pairs are considered, the embeddings collapse into a lower dimensional space, which also commonly occurs in self-supervised learning (Jing et al., 2022). However, as the dimensions approach collapse, the dissimilarity term also approaches a dimension-mean regularizer. Our findings show that while the dissimilarity term in the SG loss is not itself a dimension regularizer when the term is most needed to counteract the similarity attraction, dissimilarity preservation can be achieved via regularization.

We operationalize the dimension-based approach with an algorithm augmentation. We augment existing algorithms using SGNS by making two modifications. First, the augmentation prioritizes similarity preservation over dissimilarity preservation. This is desirable because, in real-world graph data, the lack of similarity between two nodes does not necessarily suggest the two nodes are dissimilar; it is also possible that data are missing or noisy (Newman, 2018; Young et al., 2021). Second, when the embeddings begin to collapse after optimizing only for similarity preservation for a fixed number of epochs, our augmentation repels nodes from each other using a regularizer that induces embedding dimensions centered on the origin.

In summary, our contributions are as follows:

1. In Section 2, we introduce a framework that maps node repulsion to dimension regularization. We show that instead of shortcutting the full skip-gram loss function with SGNS and repelling a sample of pairs, the repulsion function can be approximated with a dimension-mean regularization. We prove that as the need for node repulsion grows, optimizing the regularizer converges to optimizing the skip-gram loss. This framework extends the equivalence between sample-contrastive objectives and dimension regularization established in self-supervised learning.

2. In Section 3 we introduce a generic algorithm augmentation that prioritizes node attraction and replaces SGNS with occasional dimension regularization for any existing SG algorithm. We instantiate the augmentation for node2vec and LINE, reducing the repulsion complexity from $\mathcal{O}(n)$ to $\mathcal{O}(d)$ per epoch.

3. In Section 4 we show that our augmentation reduces runtime and memory usage while also preserving link-prediction performance. Moreover, in sparse networks with high local density, removing repulsion altogether, a special case of our framework, even improves performance; however, more generally, in networks with low local density, repulsion is needed and dimension regularization provides an efficient solution.

## 2 FROM NODE REPULSION TO DIMENSION REGULARIZATION

In this section, we introduce our loss decomposition framework where a function $P$ operationalizes similarity preservation and a "negative" function $N$ achieves dissimilarity preservation. We then

Table 1: Notations used in this paper.

| Symbol | Meaning |
|---|---|
| $G, V, E$ | Graph $G$ with vertices $V$ and edges $E$ |
| $n, m$ | number of nodes and edges respectively |
| $d$ | number of embedding dimensions |
| $N, P$ | negative and positive loss functions |
| $S$ | similarity matrix $\in \mathbb{R}^{n \times n}$ |
| $X$ | node embedding matrix $\in \mathbb{R}^{n \times d}$ |
| $X_i$ | $i^{th}$ row of $X$, as a column vector |
| $X_{\cdot j}$ | $j^{th}$ column of $X$, as a column vector |
| $P_\alpha$ | Probability distribution with parameter $\alpha$ |
| $k$ | number of negative samples per positive pair |
| $b$ | number of positive pairs per node |
| $\eta$ | learning rate |
| $D_x$ | diagonal matrix where $x$ is the diagonal |
| $\mathcal{C}$ | the constriction of the embeddings (Def. 2.1) |
| $\vec{1}, \mathbf{1}$ | a vector and matrix of all ones, respectively. |

show that instead of optimizing negative functions with costly node repulsions, we can instead regularize dimensions. Crucially, in Subsection 2.2, we show that when node repulsion is needed, the negative function in the skip-gram loss can be optimized via dimension regularization.

Using notation introduced in Table 1, the decomposition is as follows: given an embedding matrix $X \in \mathbb{R}^{n \times d}$ and a similarity matrix $S \in \{0, 1\}^{n \times n}$, where $S_{ij} = 1$ if nodes $i$ and $j$ are similar, a generic graph embedding loss function $L(X, S)$ can be written as:

$$L(X, S) = P(X, S) + N(X, S). \tag{1}$$

The decomposition in equation 1 applies to nearly all unsupervised graph embedding objectives as well as many supervised learning objectives, where supervision is provided in the form of node labels. In the recent graph embedding survey by Chami et al. (2022), the decomposition applies to all unsupervised methods except for Graph Factorization (Ahmed et al., 2013), which does not include a negative function $N$. Examples of popular decomposable loss functions are matrix reconstruction error (e.g., spectral embeddings) as well as softmax (e.g. node2vec (Grover & Leskovec, 2016) and LINE (Tang et al., 2015)). The decomposition also applies to supervised methods that regularize for graph structure ($\beta > 0$ as defined in Chami et al. (2022)), such as Neural Graph Machines (Bui et al., 2018) and Planetoid (Yang et al., 2016).

Given that graphs are sparse, performing gradient descent on $N$ is costly as $\nabla N$ repels all dissimilar pairs, resulting in $\mathcal{O}\left(n^2\right)$ vector additions per epoch. In this paper, we build on the argument that the costly *node-wise* operation can be replaced with a more efficient *dimension-wise* operation. Optimizing from the dimension perspective also yields a simpler geometric interpretation. This interpretation is illustrated in Figure 1.

Below, we will map the two dominant negative functions found in graph embedding algorithms to dimension regularizations. The two loss functions are spectral loss functions and skip-gram loss functions. The mapping of spectral loss functions to dimension covariance regularization parallels recent approaches to non-contrastive learning (Bardes et al., 2022; Garrido et al., 2022). Our novel contribution is a mapping from the skip-gram loss to a regularizer that induces origin-centered dimensions. Proof for all propositions below are included in Appendix A.

## 2.1 Dimension Regularization for Spectral Embeddings

In the case of Adjacency Spectral Embeddings (ASE) (Daniel L. Sussman & Priebe, 2012), which are equivalent to taking the leading eigenvectors of the adjacency matrix, the matrix $S$ is the adjacency matrix $A \in \{0, 1\}^{n \times n}$. For convenience, we define $P$ and $N$ for an individual node $i$, where

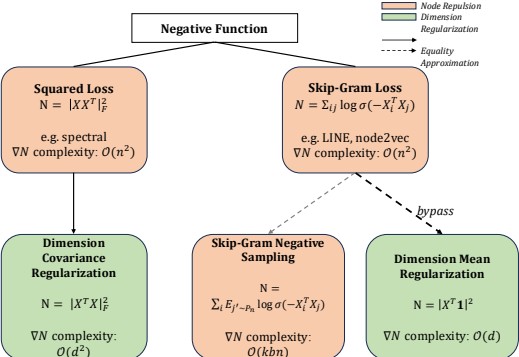

Figure 1: Nearly all unsupervised and many supervised graph embedding loss functions define a "negative" function that repels embeddings of dissimilar nodes. We show that instead of repelling pairs of nodes (orange), which is costly, the negative function in the popular skip-gram (SG) loss can be approximated with a dimension-mean regularizer. The regularizer is efficient given that $d \ll n$. This result complements the duality between squared-loss and dimension covariance utilized in self-supervised learning. Runtime complexities are expressed in terms of the number of vector additions per epoch with details in Section 2.2.1.

the full function simply sums over all nodes (e.g., $L(X, S) = \sum_{i \in V} L(X, S, i)$):

$$L_{ASE}(X, S, i) = \|S_i - X_i X^T\|_2^2, \tag{2}$$

$$P_{ASE}(X, S, i) = -2 \sum_{j \in \{j' | S_{ij'} = 1\}} X_i^T X_j + \|S\|_F^2, \tag{3}$$

$$N_{ASE}(X, S, i) = \|X X_i\|_2^2. \tag{4}$$

On one hand, performing gradient descent on $N_{ASE}$ can be interpreted as repelling all pairs of embeddings where the repulsion magnitude is the dot product between embeddings. If $\eta$ is a learning rate and $t$ is the step count, the embedding for node $i$ is updated as:

$$X_i^{t+1} = X_i^t - 2\eta \sum_{i' \in V} \left( X_i^T X_{i'} \right) X_{i'} \quad \forall i \in V. \tag{5}$$

The same negative function can also be written as a dimension covariance regularization:

**Proposition 2.1.** $N_{ASE}$ is equivalent to the regularization function $\|X^T X\|_F^2$ which penalizes covariance among dimensions.

With Proposition 2.1, we can re-interpret the gradient descent updates in equation 5 as collectively repelling dimensions. The gradient update can now be written in terms of dimensions:

$$X_{\cdot j}^{t+1} = X_{\cdot j}^t - 2\eta \sum_{j' \in [d]} \left( X_{\cdot j}^T X_{\cdot j'} \right) X_{\cdot j'} \quad \forall j \in [d]. \tag{6}$$

## 2.2 DIMENSION REGULARIZATION FOR SKIP-GRAM EMBEDDINGS

We now introduce a dimension-based approach for skip-gram embeddings. For skip-gram embeddings, the similarity matrix is defined such that $S_{ij} = 1$ if node $j$ is in the *neighborhood* of $i$. For first-order LINE, the neighborhood for node $i$ is simply all nodes connected to $i$ whereas for node2vec, the neighborhood is defined as all nodes within the context of $i$ on a random walk. The

skip-gram (SG) loss functions can be decomposed as:

$$L_{SG}(X, S, i) = - \sum_{j \in V} S_{ij} \log \sigma \left( X_i^T X_j \right) + (1 - S_{ij}) \log \sigma \left( -X_i^T X_j \right), \tag{7}$$

$$P_{SG}(X, S, i) = - \sum_{j \in \{j' | S_{ij'} = 1\}} \log \sigma \left( X_i^T X_j \right), \tag{8}$$

$$N_{SG}(X, S, i) = - \sum_{j \in \{j' | S_{ij'} = 0\}} \log \sigma \left( -X_i^T X_j \right). \tag{9}$$

Our goal is to map $N_{SG}$ to a dimension regularization. Recall that this work is motivated by the fact that the purpose of $N_{SG}$ is to prevent the similarity $\sigma \left( X_i^T X_j \right)$ from increasing for all $i, j$; without $N_{SG}$, trivial embedding solutions can emerge that maximize similarity for all pairs of nodes, not just similar pairs.

**Guaranteed collapse.** To measure the onset of the degenerate condition in which all pairs of nodes are similar, we define the constriction $\mathcal{C}$ of a set of embeddings to be the minimum dot product between any pair of nodes:

**Definition 2.1** (Constriction). The constriction $\mathcal{C}$ of an embedding matrix $X$ is defined as: $\mathcal{C} = \min_{i,j \in [n] \times [n]} X_i^T X_j$.

Geometrically, the embedding constriction is maximized when embeddings are radially squeezed and growing in magnitude, that is, collapsed. Proposition 2.2 states that if we remove $N_{SG}$ altogether and the embeddings are initialized with sufficiently small norm and learning rate, the degenerate collapse will inevitably arise during the course of gradient descent. In the context of graph neural networks, Proposition 2.2 provides conditions under which embedding oversmoothing is guaranteed. In Appendix A.2.1, we validate Proposition 2.2 by showing that embeddings inevitably collapse if only attraction updates are applied for various synthetic sparse graphs.

**Proposition 2.2.** As the Euclidean norm of the initial embeddings and the learning rate approach zero, then in the process of performing gradient descent on $P_{SG}$, there exists a step $t$ such that for all gradient updates after $t$, the constriction $\mathcal{C}$ is positive and monotonically increasing.

The proof sketch for Proposition 2.2 is as follows: as the embeddings are initialized closer to the origin, gradient descent on $P_{SG}$ approaches gradient descent on the matrix completion loss function: $\|\mathbf{1}_{S>0} \odot \left( \mathbf{1} - \frac{1}{2} XX^T \right) \|_F^2$, where $\mathbf{1}_{S>0}$ is the indicator matrix for whether entries of $S$ are positive. From Gunasekar et al. (2017), gradient descent implicitly regularizes matrix completion to converge to the minimum nuclear solution; this implicit regularization drives all dot products to be positive, not just pairs of embeddings corresponding to connected nodes.

**Approaching dimension regularization.** Now, we show that as constriction increases, performing gradient descent on $N_{SG}$ approaches optimizing a dimension regularizer. That is, when repulsion is most needed and the embeddings approach collapse due to similarity preservation, repulsion can be achieved via regularization.

First, we map $N_{SG}$ to an "all-to-all" node repulsion. While $N_{SG}$ only sums over negative node pairs ($i, j$ where $j$ is *not* in the neighborhood of $i$), for large, sparse graphs we can approximate $N_{SG}$ with the objective $N'_{SG}$ which sums over all pairs of nodes:

$$N'_{SG} = - \sum_{i,j} \log \sigma \left( -X_i^T X_j \right). \tag{10}$$

Proposition 2.3 states that if the embedding norms are bounded and the constriction $\mathcal{C} > 0$, then, in the limit of $n$, the difference between the gradient of $N'_{SG}$ and $N_{SG}$ approaches zero.

**Proposition 2.3.** If all embeddings have norms that are neither infinitely large or vanishingly small and the embedding constriction $\mathcal{C} > 0$, then, as the number of nodes in a sparse graph grows to infinity, the gradients of $\nabla N_{SG}$ and $\nabla N'_{SG}$ converge:

$$\lim_{n \to \infty} \frac{\|\nabla N'_{SG} - \nabla N_{SG}\|_F^2}{\|\nabla N_{SG}\|_F^2} = 0, \tag{11}$$

where a graph is sparse if $|E|$ is $o(n^2)$.

For a single node $i$, performing gradient descent on equation 10 results in the following update:

$$X_i^{t+1} = \left(1 - \eta\sigma\left(\|X_i^t\|^2\right)\right) X_i^t - \eta \sum_{i' \in V} \sigma\left(\left(X_i^t\right)^T X_{i'}^t\right) X_{i'}^t. \tag{12}$$

The right-hand term in the gradient update repels node $i$ from all other nodes where the repulsion is proportional to the similarity between the node embeddings. In Proposition 2.4, we show a connection between minimizing $N_{SG}'$ and centering the dimensions at the origin. For intuition, observe that if all pairs of nodes are highly similar, the gradient update in equation 12 is approximately equal to subtracting the column means scaled by a constant $(2\eta\left(X^T\vec{1}\right))$. This is equivalent to performing gradient descent on a dimension regularizer that penalizes non-zero dimension means,

$$R(X) = \|X^T\vec{1}\|_2^2. \tag{13}$$

We formalize the connection between the negative function $N_{SG}'$ and origin-centering in the following proposition:

**Proposition 2.4.** Let $R$ be the dimension regularizer defined in equation 13 that penalizes embeddings centered away from the origin and $n \gg d$. Then, as the constriction increases beyond zero, the difference between performing gradient descent on $R$ versus $N_{SG}'$ vanishes.

We note that our result establishing a connection between the skip-gram loss and origin-centered dimensions is analogous to the finding in Wang & Isola (2022) connecting the InfoNCE loss with embeddings uniformly distributed on unit hyperspheres.

### 2.2.1 COMPARISON WITH SKIP-GRAM NEGATIVE SAMPLING

Skip-gram negative sampling (SGNS) offers an efficient stochastic approximation to the gradient update in equation 12. Furthermore, the SGNS proceedure provides a tunable way to bias the gradients—via non-uniform sampling—in a manner that has been seen to empirically improve the utility of the resulting embedding in downstream tasks (Mikolov et al., 2013). Instead of repelling node $i$ from all other $n-1$ nodes, SGNS repels $i$ from a sample of $k$ nodes where the nodes are sampled according to a distribution $P_\alpha$ over all nodes, optimizing the following objective:

$$N_{SGNS}(X, S, i) = -k\mathbb{E}_{j' \sim P_\alpha}\left[\log\sigma\left(-X_i^T X_{j'}\right)\right], \tag{14}$$

where the expectation is estimated based on $k$ samples.

In aggregate, SGNS reduces the repulsion time complexity from $\mathcal{O}\left(n^2\right)$ to $\mathcal{O}\left(kbn\right)$ vector additions per epoch. $b$ is the average number of positive pairs per node; for LINE, $b = m/n$, and for node2vec, $b$ is the product of the context size and random-walk length. With Proposition 2.4, we reduce the time complexity to $\mathcal{O}(d)$ vector additions per epoch.

As mentioned, SGNS embeddings can be tuned by the choice of the non-uniform sampling distribution, where in graph embedding contexts the distribution $P_\alpha$ is typically sampling nodes proportional to their degree$^\alpha$, with $\alpha = 3/4$. An optimization-based intuition for this choice is that a degree-based non-uniform distribution prioritizes learning the embeddings of high-degree nodes, but we emphasize that the specific choice of $\alpha = 3/4$ is typically motivated directly based on improved empirical performance in downstream tasks.

We briefly note that our dimension regularization framework is immediately amenable to introducing an analogous tuning opportunity. We can simply replace the regularization in equation 15 with

$$R(X; \vec{p}) = \|X^T\vec{p}\|_2^2, \tag{15}$$

where $\vec{p}$ is a normalized weight vector that biases the negative update in exact correspondence to the probabilities of each node in $P_\alpha$. In our later simulations, we focus our efforts on the uniform case of $\vec{p} = \vec{1}$, i.e., the regularizer in equation 15.

## 3 ALGORITHM AUGMENTATION TO REPLACE SGNS

We now propose a generic algorithm augmentation framework that directly replaces SGNS with dimension regularization. We instantiate this algorithm augmentation for LINE and node2vec but note that the framework is applicable to any graph embedding algorithm using SGNS.

Our augmentation modifies existing algorithms using SGNS in two ways. First, the augmentation prioritizes the positive function $P$, that is preserving similarity when possible. Not only does prioritizing observed edges increase efficiency, but it is also desirable given the fact that real-world graph data frequently have missing edges (Newman, 2018; Young et al., 2021). We prioritize the observed edges by defaulting to performing gradient descent on $P$.

Second, our augmentation achieves repulsion via gradient descent on $R$, the dimension-mean regularizer introduced in equation 13. As constriction increases from optimizing only $P$, dimension-mean regularization approximates $\nabla N'_{SG}$ as established in Proposition 2.4.

Taken together, the algorithm augmentation framework can be summarized as:

$$X^{t+1} = \begin{cases} X^t - \eta \nabla P_{SG}\left(X^t\right) & t \,\%\, n_{\text{negative}} \neq 0, \\ X^t - \eta \left[\nabla P_{SG}\left(X^t\right) + \lambda \nabla R\left(X^t\right)\right] & t \,\%\, n_{\text{negative}} = 0, \end{cases} \tag{16}$$

where $\lambda$ is the regularization hyperparameter, $n_{\text{negative}}$ controls the frequency of performing gradient descent on $R$, and $\eta$ is the learning rate.

## 3.1 INSTANTIATION FOR LINE AND NODE2VEC

In Algorithm 1 we include the pseudo-code for the augmented versions of augmented LINE and node2vec, utilizing the framework in equation 16. The loop labeled "old negative update" shows where SGNS would have occurred.

---

**Algorithm 1** Augmented LINE and node2vec

---

**Input:** $G, n, d, p, q, \text{num\_batches}, \text{batch\_size}, \lambda, \eta, n_{\text{negative}}$
$X^0 \leftarrow \text{random\_initialization}(n, d)$
$walks \leftarrow \text{run\_random\_walks}(G, p, q)$
**for** $t \in \{1, \dots, \text{num\_batches}\}$ **do**
    $X^{t+1} \leftarrow X^t$
    **for** $\{1 \cdots \text{batch\_size}\}$ **do**
        $i, j \leftarrow \text{sample\_uniform\_pair}(walks)$
        $X_i^{t+1} \mathrel{+}= \eta\sigma\left(-\langle X_i^t, X_j^t \rangle\right) X_j^t$                          $\triangleright$ positive update
        $X_j^{t+1} \mathrel{+}= \eta\sigma\left(-\langle X_i^t, X_j^t \rangle\right) X_i^t$
        **for** $j' \in \text{sample}(P_\alpha, k)$ **do**            $\triangleright$ old negative update (removed)
            $X_i^{t+1} \mathrel{-}= \eta\sigma\left(-\langle X_i^t, X_{j'}^t \rangle\right) X_{j'}^t$
            $X_{j'}^{t+1} \mathrel{-}= \eta\sigma\left(-\langle X_i^t, X_{j'}^t \rangle\right) X_i^t$
        **end for**
    **end for**
    **if** $t \,\%\, n_{\text{negative}} == 0$ **then**                     $\triangleright$ new negative update
        $X^{t+1} \mathrel{-}= \frac{\lambda}{n}\mathbf{1}X^t$
    **end if**
**end for**

---

LINE and node2vec differ in their implementation of the random-walk generation function. For LINE, the function simply returns the edge set $E$. For node2vec, the function returns the set of all node and neighbor pairs $(\{i, j | S_{ij} = 1\})$ where neighbors of $i$ are nodes encountered on a biased random walk starting at $i$. The parameters $p, q$ control the bias of the random walk, as specified in Grover & Leskovec (2016).

## 4 EXPERIMENTS

To assess the efficacy of our dimension-regularization augmentation for capturing topological structure, we evaluate the link-prediction performance of the node2vec and LINE instantiations. Our results show that replacing SGNS with dimension regularization preserves link-prediction performance while reducing both training time and GPU memory usage. In particular, for real-world networks, we show that applying no repulsion at all, a special case of our framework, nearly always

outperforms LINE. We correlate the success of attraction-only models with local *density*, which is common in real-world networks. However, more generally, in networks with local *sparsity*, repulsion is needed, and dimension regularization provides an efficient alternative to SGNS. All of the code for the below experiments can be found in the Supplemental Materials.

## 4.1 METHODOLOGY

We utilize the popular link-prediction benchmarks of Cora, CiteSeer, and PubMed (Yang et al., 2016) as well as three OGB datasets: ogbl-collab, ogbl-ppa, and ogbl-citation2 (Hu et al., 2020). We split each dataset into training, validation, and testing edge sets, where the validation set is used for hyperparameter optimization. Further details on our data split, hyperparameter optimization, and compute setup are in Appendix B.

For each of node2vec and LINE, we instantiate three model variants: variant $I$ is vanilla node2vec/LINE; variant $II^0$ is a special instance of our framework in which no repulsion is applied at all ($n_{negative}$ is set to be larger than the number of batches); and variant $II$ is our augmented model in which dimension regularization is applied at least once per epoch. For completeness, we also instantiate variants of the vanilla algorithms and our augmentation in which node repulsion is proportional to degree$^\alpha$ as discussed in Section 2.2.1; these results are comparable to the results for $I$ and $II$ and included in Appendix C.

We report AUC-ROC, MRR, and Hits@$k$ evaluation metrics, where AUC-ROC captures global performance while MRR and Hits@$k$ capture node-level link-prediction performance.

## 4.2 RESULTS

**Dimension regularization reduces training time and memory while preserving performance.** Tables 2 and 3 show the training time, GPU memory usage, and AUC-ROC for vanilla and augmented node2vec and LINE. The training time is aggregated over multiple epochs, where the number of epochs is fixed for each graph and specified in Appendix B. The reported GPU memory is the maximum GPU memory allocated by PyTorch over the course of training. The $\Delta(\%)$ column reports the relative difference between $II$ and $I$.

For node2vec, training time reduces by 12.2%, on average, and memory by 30.5%, on average. Meanwhile, AUC-ROC decreased by at most 6.3%. For LINE, training time decreased by 18.9%, and on average AUC-ROC increased by 4.5%. Our augmentation did not reduce memory usage for LINE because we obtained the best performance across all models when using small batch sizes; with small batches, even the vanilla algorithm has low GPU memory usage and negative sampling contributes minorly to memory usage. The small batch size also explains the increase in training time from node2vec to LINE.

Table 2: Performance of vanilla and augmented node2vec

| Dataset | Time (min) | | | | Memory (GB) | | | | AUC-ROC | | | |
|---|---|---|---|---|---|---|---|---|---|---|---|---|
| | $I$ | $II^0$ | $II$ | $\Delta(\%)$ | $I$ | $II^0$ | $II$ | $\Delta(\%)$ | $I$ | $II^0$ | $II$ | $\Delta(\%)$ |
| **CiteSeer** | 1.67 | 1.48 | 1.48 | (-11.4%) | 3.12 | 2.09 | 2.09 | (-33.0%) | 0.77 | 0.73 | 0.74 | (-3.9%) |
| **Cora** | 1.45 | 1.28 | 1.28 | (-11.7%) | 3.12 | 2.08 | 2.08 | (-33.3%) | 0.81 | 0.77 | 0.79 | (-2.5%) |
| **PubMed** | 8.33 | 6.72 | 6.73 | (-19.2%) | 3.16 | 2.12 | 2.12 | (-32.9%) | 0.79 | 0.74 | 0.74 | (-6.3%) |
| **ogbl-collab** | 9.67 | 9.03 | 8.98 | (-7.1%) | 13.25 | 9.00 | 9.00 | (-32.1%) | 0.91 | 0.81 | 0.86 | (-5.5%) |
| **ogbl-ppa** | 24.03 | 23.00 | 23.15 | (-3.7%) | 14.41 | 10.16 | 10.16 | (-29.5%) | 0.99 | 0.58 | 0.93 | (-6.1%) |
| **ogbl-citation2** | 43.27 | 33.03 | 34.78 | (-19.6%) | 19.25 | 15.00 | 15.00 | (-22.1%) | 0.79 | 0.67 | 0.82 | (3.8%) |

**In the case of globally sparse but locally dense networks, removing repulsion altogether improves performance.** Interestingly, we observe that for LINE, the special case of our algorithm augmentation that removes repulsion altogether ($II^0$) outperforms vanilla LINE for five of the six graphs. Figure 2 correlates this gain in performance with local edge density, as measured by the average clustering coefficient. The trend is most noticeable in the case of ogbl-collab which has the highest clustering coefficient and reflects increases in MRR and Hits@$k$ when repulsion is removed.

Intuitively, removing repulsion is effective in globally sparse but locally dense graphs, which characterizes many real-world networks (Newman, 2003), because the global sparsity hinders dimensional

Table 3: Performance of vanilla and augmented LINE

| Dataset | Time (min) | | | | Memory (GB) | | | | AUC-ROC | | | |
|---|---|---|---|---|---|---|---|---|---|---|---|---|
| | I | II$^0$ | II | $\Delta$(%) | I | II$^0$ | II | $\Delta$(%) | I | II$^0$ | II | $\Delta$(%) |
| CiteSeer | 14.15 | 12.00 | 11.78 | (-16.7%) | 0.02 | 0.02 | 0.02 | (0.00%) | 0.57 | 0.63 | 0.59 | (3.5%) |
| Cora | 16.50 | 13.82 | 13.58 | (-17.7%) | 0.02 | 0.02 | 0.02 | (0.00%) | 0.56 | 0.63 | 0.53 | (-5.4%) |
| PubMed | 5.23 | 4.48 | 4.48 | (-14.3%) | 0.06 | 0.06 | 0.06 | (0.00%) | 0.59 | 0.65 | 0.67 | (13.6%) |
| ogbl-collab | 43.28 | 34.17 | 34.10 | (-21.2%) | 0.63 | 0.63 | 0.63 | (0.00%) | 0.63 | 0.69 | 0.63 | (0.00%) |
| ogbl-ppa | 64.57 | 51.50 | 51.05 | (-20.9%) | 1.97 | 1.97 | 1.97 | (0.00%) | 0.88 | 0.94 | 0.92 | (4.5%) |
| ogbl-citation2 | 64.93 | 50.65 | 50.55 | (-22.1%) | 8.02 | 8.02 | 8.02 | (0.00%) | 0.57 | 0.66 | 0.63 | (10.5%) |

collapse while the local density ensures that node attraction brings nodes near hidden neighbors. While we would theoretically expect the embeddings to collapse even in sparse networks, in our experiments, we use the Adam optimizer which decays the learning rate. On the other hand, removing repulsion is less effective for node2vec because the random walks effectively increase global density, and thus, the likelihood of embedding collapse.

**More generally, repulsion is needed and dimension regularization provides a scalable solution.** To generalize beyond the six real-world graphs, we examine the performance of our augmentation on Stochastic Block Model graphs in which we toggle local density. The right side of Figure 2 shows that when the within-block edge probability is much greater than the between-block probability, all variants perform well. However, as local density decreases and the boundary between blocks erodes, repulsion is needed as indicated by the gap between I and II$^0$; the figure shows that dimension regularization (II) is an effective bridge.

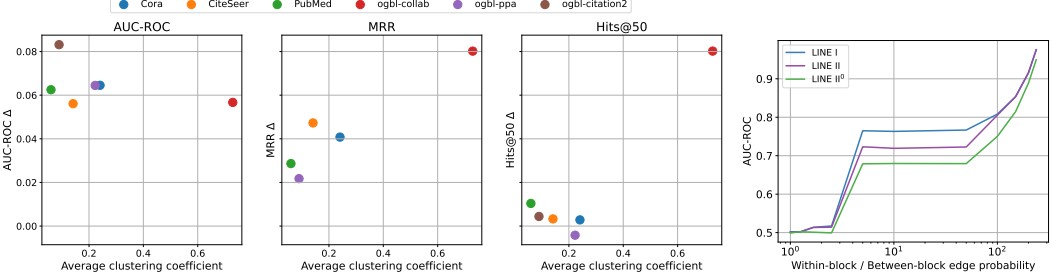

Figure 2: In practice, removing repulsion altogether performs well for real-world graphs that are globally sparse but locally dense (high clustering coefficient). On the left, we show the change in performance when repulsion is removed relative to vanilla LINE. For almost all graphs, AUC-ROC, MRR, and Hits@$k$ all increase; the increase is most prominent in the case of ogbl-collab which has high local density. However, on the right, we show that as we decrease local density in a two-block Stochastic Block Model (SBM), the need for repulsion increases and dimension regularization provides an efficient and effective repulsion mechanism.

## 5 RELATED WORKS

In this section, we review the popular use of SGNS within graph embeddings as well as its limitations. Our approach is also similar in spirit to the growing body of literature on non-contrastive learning.

### 5.1 SKIP-GRAM NEGATIVE SAMPLING

SGNS was introduced in word2vec by Mikolov et al. (2013) as an efficient method for learning word embeddings. While the softmax normalization constant is costly to optimize, Mikolov et al. (2013) modeled SGNS after Noise Contrastive Estimation (NCE) which learns to separate positive samples from samples drawn from a noise distribution. SGNS has since been adopted for graph representation learning where it is utilized in both unsupervised (Grover & Leskovec, 2016; Tang et al., 2015; Perozzi et al., 2014) and supervised skip-gram models (Yang et al., 2016).

At the same time, there are many known limitations of SGNS. Rudolph et al. (2016) place SGNS embeddings within the framework of Exponential Family Embeddings and note that SGNS down-weights the magnitude of the negative update and leads to biased embeddings, relative the gradients of the non-sampled objective. Second, Davison & Austern (2023) examine the limiting distribution of embeddings learned via SGNS and show that the distribution decouples from the true sampling distribution in the limit. Last, it has also been shown that the optimal noise distribution and the corresponding parameters can vary by dataset (Yang et al., 2020).

We would also like to note that while the motivations are similar, SGNS differs from the negative sampling that has arisen in the self-supervised learning literature (Robinson et al., 2021). In self-supervised learning, the negative samples are generally other nodes in the training batch.

## 5.2 Non-contrastive Self-Supervised Learning

Energy Based Models in self-supervised learning are a unified framework for balancing similarity and dissimilarity (LeCun et al., 2007). As in our decomposition, energy-based models ensure that similar pairs have low energy and dissimilar pairs have high energy. Within energy-based models, there has been more focus across both vision and graph representation learning on contrastive models, which explicitly repel dissimilar pairs (In et al., 2023; Li et al., 2023; Yang et al., 2023; Zhang et al., 2023). However, given the computational complexities of pairwise contrast, there is a growing body of work on non-contrastive representation learning methods, which do not use negative samples. Embedding collapse is the main challenge facing non-contrastive methods, and several mitigation strategies have emerged such as asymmetric encoders in SimSiam (Chen & He, 2021), momentum encoders in BYOL (Grill et al., 2020), and redundancy reduction in Barlow Twins (Zbontar et al., 2021). Garrido et al. (2022) establishes a duality between dimension-regularization based non-contrastive approaches and standard contrastive learning; however, the work specifically analyzes the squared-loss term and there are no existing works, to our knowledge, establishing a connection between skip-gram loss and non-contrastive methods.

## 6 CONCLUSION

In this work, we provide a new perspective on dissimilarity preservation in graph representation learning and show that dissimilarity preservation can be achieved via dimension regularization. Our main theoretical finding shows that when node repulsion is most needed and embedding dot products are all increasing, the difference between the original skip-gram dissimilarity loss and the dimension-mean regularizer vanishes. Combined with the efficiency of dimension operations over node repulsions, dimension regularization bypasses the need for SGNS. We then introduce a generic algorithm augmentation that prioritizes positive updates, given that real-world graph data often contain missing edges Young et al. (2021). When node repulsion is needed and collapse approaches, the augmentation utilizes dimension regularization instead of SGNS. Our experimental results show that the augmented versions of LINE and node2vec preserve the link-prediction performance of the original algorithms while reducing runtime by over 30% for OGB benchmark datasets. In fact, for several real-world graphs, removing dissimilarity preservation altogether performs well; however, more generally, for graphs with low local density repulsion is needed, and dimension regularization is an efficient and effective approach.

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

## A    PROOFS

### A.1    PROOF FOR PROPOSITION 2.1

*Proof.* Recall that the Frobenius norm of a matrix is equivalent to the trace of the corresponding Gram matrix:

$$N_{ASE}(X, S) = \|XX^T\|_F^2 \tag{17}$$

$$= \text{Tr}\left(XX^T \left(XX^T\right)^T\right) \tag{18}$$

$$= \text{Tr}\left(XX^T XX^T\right) \tag{19}$$

$$= \text{Tr}\left(X^T XX^T X\right) \tag{20}$$

$$= \text{Tr}\left(\left(X^T X\right)\left(X^T X\right)^T\right) \tag{21}$$

$$= \|X^T X\|_F^2, \tag{22}$$

$\square$

where the 4th line follows from the cyclic property of the trace.

### A.2    PROPOSITION 2.2

*Proof.* From gradient descent, we know that $X_i^T X_j$ increases toward infinity for all $i, j \in E$; however, to show that $X_i^T X_j$ increases for *all* $i, j$, we show that the cosine similarity for all pairs of embeddings approaches 1. We characterize the embedding dynamics in two phases: in the first phase (alignment), the embeddings are initialized near the origin and then converge in direction; then, in the second phase (asymptotic), the embeddings asymptotically move away from the origin while maintaining alignment.

*Phase 1: alignment.* The gradient update rule for $P_{SG}$ is:

$$X_i^{t+1} = X_i^t + \eta \sum_{j \in N(i)} \sigma\left(-(X_i^t)^T X_j^t\right) X_j^t \tag{23}$$

Because the embeddings are initialized sufficiently small, the sigmoid function can be approximated linearly via a first-order Taylor expansion:

$$\sigma(z) \approx \frac{1}{2} + \frac{1}{4}z \tag{24}$$

In this case, the update rule becomes:

$$X_i^{t+1} \approx X_i^t + \frac{\eta}{2} \sum_{j \in N(i)} \left(1 - \frac{1}{2}\left((X_i^t)^T X_j^t\right)\right) X_j^t \tag{25}$$

The above gradient is equivalent to performing gradient descent on:

$$P'_{SG} = \|\mathbf{1}_{S>0} \odot \left(\mathbf{1} - \frac{1}{2}XX^T\right)\|_F^2 \tag{26}$$

From Gunasekar et al. (2017), gradient descent for matrix completion is implicitly regularized to yield the minimum nuclear norm (lowest rank) stationary point. In the case where $G$ is connected, minimizing the nuclear norm implies that gradient descent on equation 26 causes $XX^T$ to approach $2\mathbf{1}$, and thus the dot products between all pairs of embeddings increases and $\mathcal{C} \to 2$.

*Phase 2: Asymptotic.* To complete the proof we show that once $\mathcal{C} > 0$, the constriction monotonically increases with each gradient-descent update to $P_{SG}$. For any pair of embeddings $X_i^t$ and $X_j^t$,

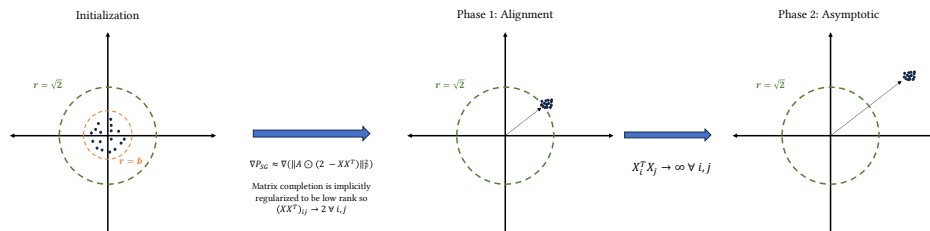

Figure 3: High-level overview of proof for Proposition 2.2, which guarantees embedding collapse when only attraction updates are applied. In the beginning the embeddings are initialized near the origin, all with norm at most $b$. Then, in Phase 1, attraction update rule is approximately gradient descent for matrix completion; given that the latter is implicitly regularized to yield low-rank solutions the embeddings converge in direction. In the second phase, the embeddings asymptotically distance away from the origin in the same direction.

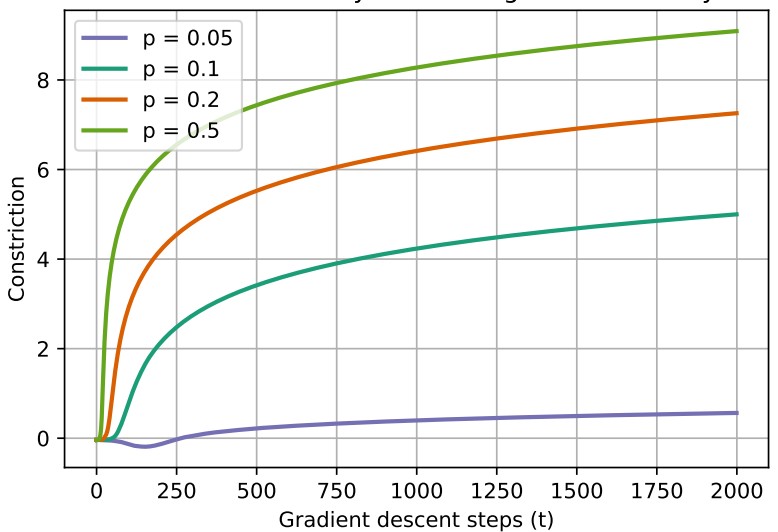

Figure 4: To empirically validate Proposition 2.2, we instantiate Erdös-Rényi networks ($n = 100$) and learn embeddings by only applying the attraction update. The figure shows that for various graph densities, the constriction eventually becomes monotonically increasing.

the dot product after a single gradient descent epoch is:

$$
\left\langle X_i^{t+1}, X_j^{t+1} \right\rangle = \left( X_i^t + \eta \sum_{k \in N(i)} \sigma\left( -(X_i^t)^T X_k^t \right) X_k^t \right)^T
$$
$$
\left( X_j^t + \eta \sum_{k \in N(j)} \sigma\left( -(X_j^t)^T X_k^t \right) X_k^t \right)
\tag{27}
$$

If $\mathcal{C} > 0$ at $t$, then all of the dot-product terms from distributing the right-hand side of equation 27 are positive, and we have $\left\langle X_i^{t+1}, X_j^{t+1} \right\rangle = \left\langle X_i^t, X_j^t \right\rangle + \delta$, where $\delta > 0$. Thus the dot product between any pair of embeddings strictly increases from step $t$ to $t + 1$ and hence the constriction monotonically increases if $\mathcal{C} > 0$ at step $t$. $\qquad\square$

### A.2.1 SUPPLEMENTAL FIGURES

Figure 3 provides a high-level summary of the proof for Proposition 2.2. In the beginning, the embeddings are initialized with norm $\leq b \leq \sqrt{2}$. Then, the embeddings converge in direction given that near the origin, the gradient of $P_{SG}$ is approximately the gradient of a matrix completion problem. Further, gradient descent for matrix completion near the origin is implicitly regularized to yield the lowest rank solution, hence a convergence in the embedding direction. Thereafter, once the dot products between all pairs of nodes are positive in the "Asymptotic" phase, $\mathcal{C}$ becomes monotonically increasing.

We also empirically validate Proposition 2.2 in Figure 4. We initialize Erdös-Rényi graphs ($n = 100$) of various densities, randomly initialize the embeddings around the origin, and then apply the positive-only update rule in equation 23. Figure 4 shows that even when the edge density is $0.05$, the constriction eventually becomes monotonically increasing.

### A.3 PROOF FOR PROPOSITION 2.3

*Proof.* Let us define the matrix of embedding similarities as $K = \sigma\left(XX^T\right)$. Then, the gradient of $N'_{SG}$ is:

$$\nabla N'_{SG} = 2KX \tag{28}$$

If $\mathbf{1}_{S==0}$ is the indicator matrix where entry $i, j$ is one if $S_{ij} = 0$, then the gradient of $\nabla N_{SG}$ is:

$$\nabla N_{SG} = \left(\mathbf{1}_{S==0} \odot K\right)X \tag{29}$$

The numerator in the proposition can be upper bounded as:

$$\|\nabla N'_{SG} - \nabla N_{SG}\|_F^2 = \sum_{i=1}^{n}\left\|\sum_{j' \in \{j|S_{ij}>0\}} \sigma\left(X_i^T X_{j'}\right)X_{j'}\right\|_2^2 \tag{30}$$

$$\leq \sum_{i=1}^{n}|\{j|S_{ij} > 0\}|\beta_{\max} \tag{31}$$

$$\leq m\beta_{\max} \tag{32}$$

Where in the above, $\beta_{\max}$ is a constant and the upper bound on embedding norm squared and $m$ is the number of non-zero entries in $S$.

Now we lower bound the denominator. The gradient can be expanded as:

$$\|\nabla N_{SG}\|_F^2 = \sum_{i=1}^{n}\left\|\sum_{j}^{n} \mathbf{1}_{S_{ij}==0}K_{ij}X_j\right\|_2^2 \tag{33}$$

We can lower bound the norm of the sum by replacing $X_j$ with the projection of $X_j$ onto $X_i$:

$$\|\nabla N_{SG}\|_F^2 = \sum_{i=1}^{n}\left\|\sum_{j}^{n} \mathbf{1}_{S_{ij}==0}K_{ij}X_j\right\|_2^2 \tag{34}$$

$$\geq \sum_{i=1}^{n}\left\|\sum_{j}^{n} \mathbf{1}_{S_{ij}==0}K_{ij}\left(\frac{K_{ij}}{\|X_i\|}\right)\left(\frac{X_i}{\|X_i\|}\right)\right\|_2^2 \tag{35}$$

Because the dot product between all pairs of embeddings is assumed to be positive, the norm of the sum is at most the sum of the norms:

$$\|\nabla N_{SG}\|_F^2 \geq \sum_{i=1}^{n}\sum_{j}^{n} \mathbf{1}_{S_{ij}==0}\left(\frac{K_{ij}}{\|X_i\|}\right)^4 \tag{36}$$

$$\geq \left(\frac{\mathcal{C}^2}{\beta_{\max}}\right)^2\left(n^2 - m\right) \tag{37}$$

where above $m$ is the number of non-zero entries of $S$.

Combining the bounds on the numerator and denominator together, we have:

$$\frac{\|\nabla N'_{SG} - \nabla N_{SG}\|_F^2}{\|\nabla N_{SG}\|_F^2} \leq \frac{\beta_{\max}^3}{\mathcal{C}^4} \frac{m}{n^2 - m} \tag{38}$$

The left term is a constant given the assumption on constriction and non-vanishing or infinite embedding norms. Further, because the graph is sparse ($m$ is $o\left(n^2\right)$), the second term goes to zero as $n \to \infty$. $\qquad\square$

### A.4    PROPOSITION 2.4

#### A.4.1    LEMMA FOR PROOF OF PROPOSITION 2.4

**Lemma A.1.** Call $f(x) = \log(1 + \exp(x))$. We show that both $f(x) - x \leq \exp(-x)$ and $|\nabla_x(f(x) - x)| \leq \exp(-x)$, i.e. have vanishing exponential tails.

*Proof.* First, note that $\log(x) \leq x - 1$. Since $e^{-x}(1 + e^x) = 1 + e^{-x}$, we have,

$$\log(e^{-x}(1 + e^x)) \leq (1 + e^{-x}) - 1$$
$$\log(1 + \exp(x)) - x \leq \exp(-x)$$
$$f(x) - x \leq \exp(-x),$$

where the first line applies the bound on $\log(x)$ to the equality, the second line organizes terms, and the third applies the definition of $f(x)$. Now we have $\nabla_x(f(x) - x) = \sigma(x) - 1$. For this,

$$\sigma(x) - 1 = \frac{-1}{1 + \exp(x)} \geq \frac{-1}{\exp(x)} = -\exp(-x).$$

Where the inequality follows reducing the value of the denominator. This concludes the proof. $\qquad\square$

#### A.4.2    PROOF FOR PROPOSITION 2.4

*Proof.* As in the proof for Proposition 2.3, let us define the similarity matrix $K = \sigma\left(X X^T\right)$.

The gradient for $N'_{SG}$, defined in equation 10, is:

$$\nabla_X N'_{SG} = 2KX \tag{39}$$

The constriction $\mathcal{C}$ is the minimum value of the matrix K. From lemma A.1, we know that as constriction increases, the difference between 1 and each of the entries of $K$ vanishes exponentially. Thus, there is a vanishing difference between $\frac{1}{2}\nabla_X N'_{SG}$ and $\mathbf{1}X$, where $\mathbf{1}$ is the $n \times n$ all-ones matrix. Note that $\mathbf{1}X$ is also the gradient of the dimension regularizer $R$ introduced in equation 13. Putting these together we have:

$$\left\|\frac{1}{2}\nabla_X N'_{SG} - \nabla R\right\|_2^2 = \left\|\left(\sigma\left(X X^T\right) - \mathbf{1}\right)X\right\|_2^2 \tag{40}$$

$$\leq \sum_{i=1}^{n} \left\|\sum_{j=1}^{n}\left(\sigma\left(X_i^T X_j\right) - 1\right)X_j\right\|^2 \tag{41}$$

$$\leq \sum_{i=1}^{n}\left(\sum_{j=1}^{n}\left(\sigma\left(X_i^T X_j\right) - 1\right)^2\|X_j\|^2\right) \tag{42}$$

$$\leq \left(\frac{n}{e^{\mathcal{C}}}\right)^2 \beta_{\max} \tag{43}$$

In the above, $\beta_{\max}$ is a constant and corresponds to the maximum embedding norm among all embeddings in $X$.

Thus, as constriction $\mathcal{C}$ increases, the difference between the gradient of $\frac{1}{2}N'_{SG}$ and $R$ vanishes exponentially. By extension, the difference between gradient descent on $N'_{SG}$ with a step-size of $\eta$ and gradient descent on $R$ with a step-size of $2\eta$ vanishes with increasing $\mathcal{C}$. $\qquad\square$

## B  EXPERIMENTAL METHODOLOGY

**Data Splitting.**  We split each dataset into training/validation/test edge splits.  For the OGB datasets, we utilize the splits provided by the original dataset, and for the Planetoid datasets we create uniform random 70/10/20 edge splits.

**Hyperparameter Optimization.**  We perform the following hyperparamter optimization routine:

1. **Select optimal parameters.**  For the vanilla algorithms, we optimize over the learning rate, and for the augmented algorithms we optimize over the learning rate, $n_{\text{negative}}$, and $\lambda$. In order to search over a larger space of parameters, we only train each configuration for 4 epochs and select the configuration that yields the highest AUC-ROC on the validation edge set. The optimal hyperparameters are listed in Table 4

2. **Select optimal number of epochs.**  For each optimal hyperparameter configuration, we then determine the number of training epochs by training for the number of epochs listed in Table 4. We then select the epoch number at which the validation AUC-ROC is maximized.

3. **Evaluate.** For each graph and model variant, we train a model using the optimal hyperparameters and number epochs and evaluate on the test edge set.

**Hardware.**  All of the experiments were executed on a machine with a single NVIDIA V100 GPU.

## C  SUPPLEMENTAL EVALUATION

For completeness, include the MRR and Hits@$k$ evaluation metrics here to complement the AUC-ROC results in the main body. Figure 5 includes the metrics for LINE and Figure 6 includes the metrics for node2vec. In all of the plots, the training time is on the x-axis, and the evaluation metric is on the y-axis. The figures also include the results in which node repulsion is weighted by node degree; these are denoted by $\text{I}(\alpha = 3/4)$ and $\text{II}(\alpha = 3/4)$. Overall, these results complement the AUC-ROC results showing that dimension regularization preserves performance while reducing training time. It is worth noting that the hyperparameter optimization selects parameters based on AUC-ROC so the MRR and Hits@$k$ test-set results are not optimized; also, the MRR calculation for ogbl-ppa led to an OOM error due to the number of test-set edges.

Table 4: Optimized hyperparameters

| Graph | Model | Variant | $\eta$ | $\lambda$ | $n_{\text{negative}}$ | Steps |
|---|---|---|---|---|---|---|
| CiteSeer | LINE | I | 0.1 | - | - | 50 |
| | | II | 0.1 | 0.01 | 1000 | 50 |
| | | $II^0$ | 0.1 | 0.01 | $10^9$ | 50 |
| | node2vec | I | 0.1 | - | - | 50 |
| | | II | 0.1 | 0.1 | 10 | 50 |
| | | $II^0$ | 0.1 | 0.1 | $10^9$ | 50 |
| Cora | LINE | I | 0.1 | - | - | 50 |
| | | II | 0.1 | 1 | 1000 | 50 |
| | | $II^0$ | 0.1 | 1 | $10^9$ | 50 |
| | node2vec | I | 0.1 | - | - | 50 |
| | | II | 0.1 | 0.1 | 10 | 50 |
| | | $II^0$ | 0.1 | 0.1 | $10^9$ | 50 |
| ogbl-citation2 | LINE | I | 0.01 | - | - | 2 |
| | | II | 0.01 | $10^{-7}$ | 1000 | 2 |
| | | $II^0$ | 0.01 | $10^{-7}$ | $10^9$ | 2 |
| | node2vec | I | 0.01 | - | - | 3 |
| | | II | 0.01 | $10^{-6}$ | 1000 | 3 |
| | | $II^0$ | 0.01 | $10^{-6}$ | $10^9$ | 3 |
| ogbl-collab | LINE | I | 0.1 | - | - | 10 |
| | | II | 0.1 | $10^{-4}$ | 1000 | 10 |
| | | $II^0$ | 0.1 | 0.01 | $10^9$ | 10 |
| | node2vec | I | 0.1 | - | - | 10 |
| | | II | 0.1 | 0.01 | 100 | 10 |
| | | $II^0$ | 0.1 | 0.01 | $10^9$ | 10 |
| ogbl-ppa | LINE | I | 0.01 | - | - | 3 |
| | | II | 0.01 | $10^{-6}$ | 5 | 3 |
| | | $II^0$ | 0.01 | $10^{-4}$ | $10^9$ | 3 |
| | node2vec | I | 0.01 | - | - | 5 |
| | | II | 0.01 | 0.001 | 5 | 5 |
| | | $II^0$ | 0.01 | 0.001 | $10^9$ | 5 |
| PubMed | LINE | I | 0.1 | - | - | 50 |
| | | II | 0.1 | 0.01 | 1000 | 50 |
| | | $II^0$ | 0.1 | 0.01 | $10^9$ | 50 |
| | node2vec | I | 0.1 | - | - | 50 |
| | | II | 0.1 | 0.1 | 1000 | 50 |
| | | $II^0$ | 0.1 | 0.1 | $10^9$ | 50 |

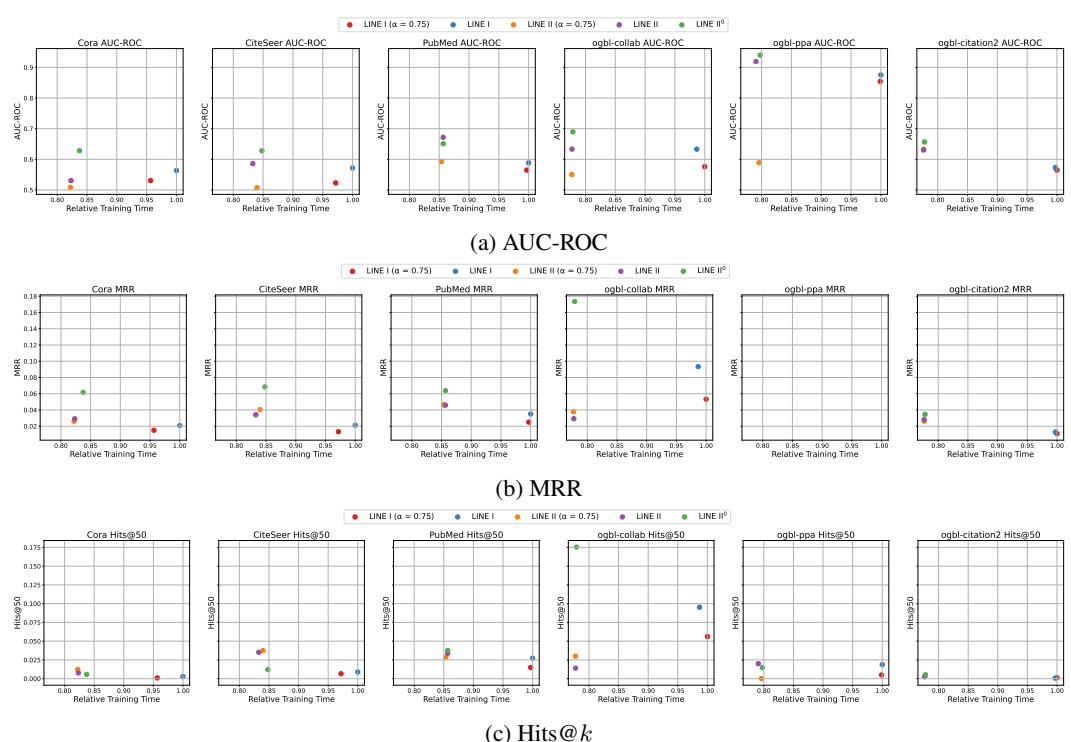

Figure 5: Performance of vanilla and augmented LINE by AUC-ROC, MRR, and Hits@$k$.

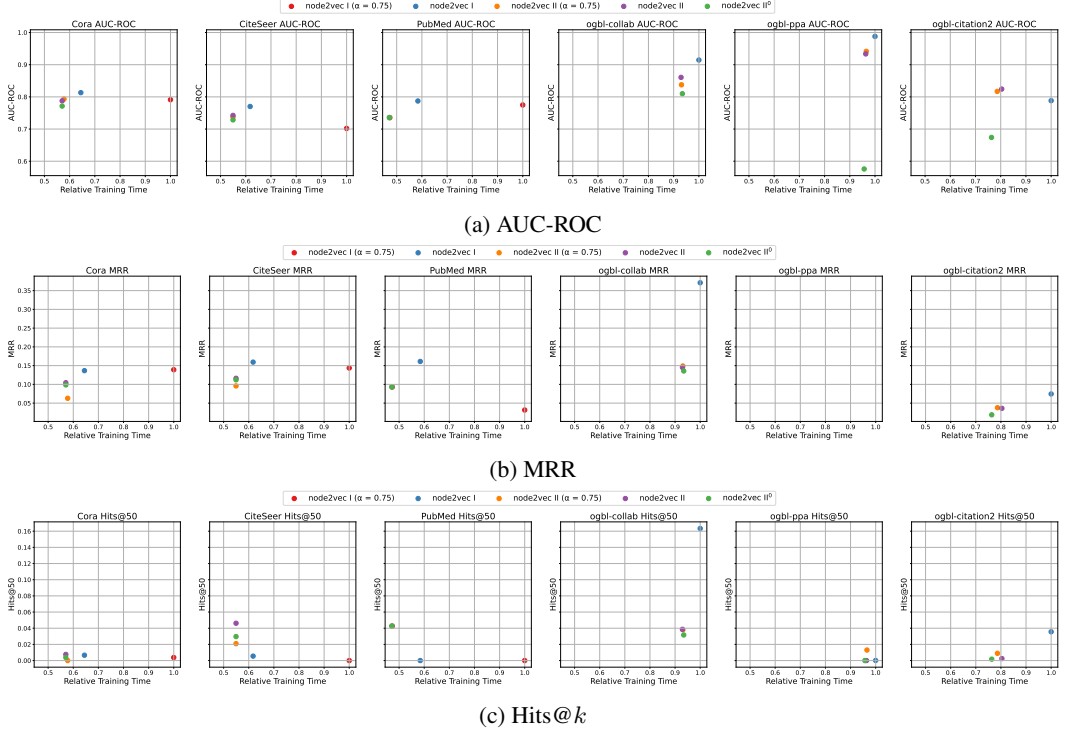

Figure 6: Performance of vanilla and augmented node2vec by AUC-ROC, MRR, and Hits@$k$.

