# OpenReview forum: "Bypassing Skip-Gram Negative Sampling: Dimension Regularization as a More Efficient Alternative for Graph Embeddings"
_ICLR.cc/2025/Conference — Submitted to ICLR 2025_

### Official Review · Reviewer_ex3Z · 2024-10-31

**Soundness:** 3
**Presentation:** 2
**Contribution:** 2
**Rating:** 5
**Confidence:** 3

**Summary:**

This paper presents a theoretical reinvestigation of skip-gram models, with particular focus on their dimensional properties. The authors demonstrate that while the skip-gram negative sampling (SGNS) loss function itself does not act as a dimension regularization technique, the preservation of dissimilarity between embeddings can be effectively achieved through dimensionality reduction methods. Building on these theoretical insights, the authors propose two key modifications to existing SGNS-based algorithms: (1) a mechanism to balance the trade-off between similarity and dissimilarity preservation, and (2) a strategic approach to introducing dimensionality regularization only after detecting embedding collapse.

**Strengths:**

1. The paper has a pretty solid mathematical interpretation (section 2). The proof flow is very good  and quite makes sense.
2. The paper is pretty well-written and easy to understand.

**Weaknesses:**

1. Limited practical impact. The proposed augmentation shows consistent performance degradation in node2vec embeddings by approximately 5% (Table 2).  The authors also fail to present potential use cases where their modifications would be beneficial for graph embedding algorithms nowadays.
2. Dataset selection is inadequate: while the key point of proposed modifications lies in potentials in scalability, the dataset selection is Limited to small and medium-scale datasets and misses evaluation on large-scale datasets.
3. Limited experiments: the experiments is mostly done on LINE and node2vec, which are pretty old graph embedding methods. It would be great if authors can provide some experiment results to justify how this can be potentially useful for  skip-gram based language models.

**Questions:**

1. Could you clarify the intended use cases for your proposed augmentation algorithm? The practical applications seem limited if the primary goal is scaling up LINE/node2vec. What other potential applications could benefit from this approach?
2. Could the proposed algorithm be used to scale up skip-gram based language models?
3. Why in table 2 and 3, the key performance metric is AUROC instead of MRR/Hits@K? Given that link prediction tasks typically rely on ranking metrics, wouldn't MRR/Hits@K be more appropriate?

---

> ### Author Response · Authors · 2024-11-25
>
> We appreciate the reviewer’s comments. Here we respond to the overall concerns regarding the practical relevance of our method as well as its novelty:
>
> The main contribution of our work is a conceptual connection between the noise contrastive estimation (NCE) loss and dimension regularization. Our connection is novel because while the SSL community has proposed dimension-based non-contrastive methods, these have been based on the squared-loss and a dimension-regularization approach has not been proposed for NCE.
>
> The main intended use case of our theoretical contribution is scaling graph embeddings to graphs where using GNNs would be prohibitive given the memory constraints. To focus on capturing network structure we evaluate on link prediction and select the largest OGB benchmark datasets available. If the reviewer has larger graphs to suggest we would be happy to evaluate on them.
>
> There are two secondary use cases for our theoretical contribution. First, our regularization is relevant in settings where graph data contain many false negatives e.g. implicit feedback. In these cases it would be better to prioritize the positive examples and not explicitly repel negative pairs, which may be false negatives. Second, another potential downstream application would be utilizing our dimension regularizer in self-supervised learning models that currently use NCE. One could eliminate negative samples in those models and utilize a  dimension-centering regularization instead.
>
> In addition, we thank the reviewer for the questions on skip-gram language models and evaluation metrics, which we answer below:
>
> > Could the proposed algorithm be used to scale up skip-gram based language models?
>
> The efficiency gain is most relevant for graph skip-gram models because the number of nodes in graphs generally exceeds the vocabulary size of language skip-gram models.
>
> > Why in table 2 and 3, the key performance metric is AUROC instead of MRR/Hits@K? Given that link prediction tasks typically rely on ranking metrics, wouldn't MRR/Hits@K be more appropriate?
>
> We include these results in the Appendix (Figures 5 and 6), and for the most part they corroborate the takeaways from the AUC-ROC results so we have omitted them from the main body to conserve space.

---

> > ### Comment · Reviewer_ex3Z · 2024-11-25
> > **Feedback  to the rebuttal**
> >
> > Thanks for your rebuttal. I will keep the score unchanged.

---

### Official Review · Reviewer_Kz8w · 2024-11-03

**Soundness:** 3
**Presentation:** 3
**Contribution:** 2
**Rating:** 5
**Confidence:** 4

**Summary:**

This paper proposes an alternate repulsive force for skipgram (instead of negative sampling) based on dimension regularization.  At its core, it takes results from self-supervised learning (dimension mean regularization) and applies them to skip gram (but mostly graph embedding).  The authors argue that the proposed negative loss is more efficient (and therefore more scalable) than skipgram with negative sampling.  Experiments to evaluate the proposed regularization are performed on a handful of mostly citation datasets.

**Strengths:**

+  The proposed method's formalization seems nice, especially in terms of how it performs in the presence of dimensional collapse.
+  The paper's writing is generally good, and the author's methods are clear.

**Weaknesses:**

-  There are some questions about novelty - in that the proposed regularization perhaps "already exists" in the SSL community and replacing SGNS is a seemingly obvious application.  However I'm not aware of work that actually does this (... but have not extensively looked for it).
- The core argument of the paper is that the proposed method is more efficient than SGNS and therefore "more scalable".   While the efficiency of the method is definitely better, its not obvious that it's an "online" loss, and the true scalability of SGNS comes from the fact that the loss can be computed (and parallelized) online.
- There are not many (or really any) baselines used in the paper's experiments.  Since skipgram is well studied, it seems like more modifications of negative sampling or other related work might aide in understanding the proposed method better.
- Finally, there is an apparent performance loss that comes from the proposed method.  This is present in the SBM experiments (Figure 2).  This is discussed some in the paper, but it seems like its the most important part and should be investigated more.

**Questions:**

Please see weaknesses.  In addition, here are some more thoughts/questions.

- What's the primary draw of the proposed method?  If its truly efficiency, then we need to see how an online version of your method performs vs SGNS.

- Using downstream tasks (especially just a handful of graphs (4 of which are citation networks)) to prove the goodness of an embedding method just doesn't cut it these days.

- I would love extra experiments in the vein of Figure 2d.  Especially, how does SGNS compare to the full gradient solution (which presumably is better than SGNS, since it is itself an efficiency play).  What can we say about when each loss is better, worse, etc.

---

> ### Author Response · Authors · 2024-11-25
>
> We thank the reviewer for their comments. Below we provide clarifications regarding the intended use case / novelty of our method along with proposals for follow up work:
>
> > What's the primary draw of the proposed method? If its truly efficiency, then we need to see how an online version of your method performs vs SGNS.
>
> The primary draw of our method is indeed efficiency, especially in cases where the memory requirements of GNNs and negative sampling in general are prohibitive. Our method unlocks the ability to utilize graph learning in settings that were previously unfeasible. A secondary draw would be establishing a conceptual connection between the noise contrastive estimation (NCE) loss and dimension regularization, specifically dimension centering.
>
> > There are some questions about novelty
>
> While the SSL community has examined non-contrastive approaches, these have focused on the squared loss which amounts to enforcing orthogonality among dimensions. To the best of our knowledge, there is not a connection between NCE loss and dimension regularization.
>
> > While the efficiency of the method is definitely better, its not obvious that it's an "online" loss.
>
> We thank the reviewer for this comment. Regarding the online setting, our augmentation inherits the online ability of SGNS. Given a stream of positive edges, our augmentation performs an attraction update for each pair and then centers the embedding dimensions at intervals of positive edges. In fact this approach is more parallelizable than SGNS for large-scale settings. For instance, in a setting where embeddings are distributed across multiple machines, the elimination of negative samples reduces the need for inter-machine communication assuming the embeddings are assigned to machines via clustering. While positive updates tend to be local, negative updates are not given that negative samples are chosen randomly across the graph.
>
> Finally, we appreciate and thank the reviewer for comments regarding additional baselines and figures similar to Figure 2d. We are running supplemental follow-up analysis evaluating our method on graphs of even larger size (where GNNs would be infeasible) and can compare against other scalability baselines apart from vanilla skip-gram models. Further, similar to Figure 2d, we are examining the performance of our model for highly sparse graphs where the positive signal is difficult to detect. In fact, per the reviewer’s comment, in these settings the class imbalance renders the full gradient suboptimal as the repulsion force dominates the attraction force.

---

> > ### Comment · Reviewer_Kz8w · 2024-11-27
> >
> > I thank the authors for their reply and look forward to the experimental results when available.

---

> > > ### Author Response · Authors · 2024-12-04
> > >
> > > We thank the reviewer for their response. We are following up regarding the hypothesis that our augmentation outperforms the baseline for heavily sparse graphs/graphs containing false-negative edges. The intuition is that standard negative sampling would incorrectly repel the false negatives. Below, we present an experiment where we modulate the relative sizes of the training and test edge sets. The results suggest that when the training set is small our augmentation outperforms the baselines. We are happy to include these results in the camera-ready.
> > >
> > > ### Setup
> > > For each of the three Planetoid datasets (CiteSeer, Cora, and PubMed), we create two train/test splits. In the first, the training set is 15% of the original edge set and in the second the training set is 75%. For each, we train the three algorithm configurations: vanilla (I), our augmentation (II), and our augmentation with no repulsion at all (II$^0$). Below, we present the link-prediction AUC-ROC and MRR metrics for LINE and node2vec.
> > >
> > > ### Results
> > >
> > > | **AUC**                  | CiteSeer (0.15) | CiteSeer (0.75) | Cora (0.15) | Cora (0.75) | PubMed (0.15) | PubMed (0.75) |
> > > |-------------------|-----------------|-----------------|-------------|-------------|---------------|---------------|
> > > | LINE I            | **0.524**           | 0.592           | 0.545       | 0.582       | **0.552**     | 0.655         |
> > > | LINE II           | 0.518           | 0.656           | 0.544       | **0.629**       | 0.546         | **0.724**     |
> > > | LINE II$^0$ | 0.521       | **0.670**       | **0.546**   | 0.622   | 0.546         | 0.705         |
> > > | node2vec I        | **0.547**       | 0.764       | **0.529**   | **0.856**   | 0.534         | **0.858**     |
> > > | node2vec II       | 0.519           | **0.775**       | 0.491       | 0.820       | **0.558**     | 0.802         |
> > > | node2vec II$^0$ | 0.521           | 0.773           | 0.499       | 0.818       | 0.544         | 0.778         |
> > >
> > >
> > > | **MRR**                  | CiteSeer (0.15) | CiteSeer (0.75) | Cora (0.15) | Cora (0.75) | PubMed (0.15) | PubMed (0.75) |
> > > |-------------------|-----------------|-----------------|-------------|-------------|---------------|---------------|
> > > | LINE I            | **0.013**       | 0.032           | 0.021       | 0.026       | 0.013     | 0.048         |
> > > | LINE II           | 0.011           | **0.036**       | 0.025       | 0.030       | **0.019**         | 0.059         |
> > > | LINE II$^0$ | 0.011           | 0.036       | **0.025**   | **0.032**   | 0.019         | **0.065**     |
> > > | node2vec I        | 0.016           | **0.113**       | 0.013       | **0.143**   | **0.048**     | **0.178**     |
> > > | node2vec II       | **0.022**       | 0.088           | **0.014**   | 0.079       | 0.040         | 0.129         |
> > > | node2vec II$^0$ | 0.022       | 0.088           | 0.014   | 0.079       | 0.041         | 0.113         |
> > >
> > > For LINE, when the training set is 75% of the edge set (the more standard split), our augmentation (II and II$^0$) always outperforms the baseline. This supports the results in the current submission.
> > >
> > > For node2vec, in cases where the vanilla baseline outperforms our augmentation on the 75% split, our augmentation outperforms the baseline on the 15% in at least one of the metrics. Specifically, for Cora (0.15), our augmentation scores higher on MRR, and for PubMed (0.15), our augmentation scores higher on AUC.
> > >
> > > We are happy to include these results in the camera-ready along with further analysis characterizing the performance improvement our augmentation offers in the sparse regime.

---

### Official Review · Reviewer_Fdqs · 2024-11-03

**Soundness:** 2
**Presentation:** 3
**Contribution:** 3
**Rating:** 5
**Confidence:** 3

**Summary:**

The paper proposes a regularization approach over embedding dimensions as an alternative to the conventional negative sampling strategy (i.e. noise contrastive estimation) adapted in the skip-gram-based graph representation learning methods. The authors apply the approach to the objectives of Node2Vec and LINE and evaluate the performance across multiple real-world datasets. The results demonstrate improvements in both computational efficiency, including reduced runtime and lower GPU memory requirements during optimization.

**Strengths:**

- Proposes a novel framework aimed at reducing running time and GPU memory requirements.
- Supports the proposed framework with theoretical analysis.

**Weaknesses:**

- Given the current dominance of Graph Neural Networks (GNNs) in the graph representation learning domain, the practical relevance of the proposed framework may be limited.
- Application of the framework to Node2Vec and LINE results in significant performance drops (notably for Node2Vec), raising concerns about its practical effectiveness.
- The theoretical analysis provided lacks detail, making some claims difficult to follow.

**Questions:**

**Comments**

**Theoretical Analysis:**
- The provided theoretical analysis lacks rigor and clarity in certain parts:

- - For instance, in the proof of Proposition 2.2, it is not clear how we obtain the equation given in Line 730 from Equation 27. Although $C$ is defined as the minimum of the inner product of embedding vectors, $C$ is considered a constant, but embeddings depend on time and vary at every step, so I think $C$ also should depend on time. In Lines 731-732, the authors state that the constriction is monotonically increasing, since it remains positive. But this argument doesn’t guarantee that we will obtain $C\geq c$ for any given $c$. The series of the reciprocals of powers of 2 is monotonically increasing as well and convergent, so it can be given as a counter-example. The learning rate is also assumed to be convergent to 0 in Proposition, but it is ignored in the proof.

- - In Lines 257-259, the authors state that we can approximate $N_{SG}$ with the objection $N_{SG}^{’}$ for large sparse graphs, and then they provide Proposition 2.3 to validate this point. However, Equation 11 holds since the denominator term goes to infinity so it does not establish the convergence of the gradients of $N_{SG}$ and $N_{SG}^{’}$
- - The derivation from Equation 37 to Equation 35 is not clearly explained.
- - A more detailed explanation in proofs and theoretical analysis would enhance the paper's quality.

**Experimental Validation:**
- Applying the proposed method to additional approaches like VERSE [1] could strengthen the experimental validation.

*[1] Tsitsulin, Anton, et al. "Verse: Versatile graph embeddings from similarity measures." Proceedings of the 2018 world wide web conference. 2018.*

**Notations and Clarity:**
- The notation in Defn. 2.1, $min_{i,j \in n x n}$ might be replaced by $min_{i,j \in [n] x [n]}$ to represent $[n]$ as a set of integers from $1$ to $n$.
- In Line 357, $j$ is the index representing batch, but it is overwritten in Line 358.
- In Equation 30, there should be a delta before $N_{SG}$.


**Questions:**
- Given that only positive samples are used (i.e. $II^0$), it is interesting that embeddings do not collapse and that models (especially LINE) achieve comparable performance to vanilla versions. Could this be related to the choice of a small number of epochs?
- Since the approach assumes unweighted edges, could the authors discuss its applicability to weighted networks?

I vote for a score of 3 because of my concerns regarding the originality of the proposed approach, theoretical contributions, and experimental evaluation.

---

> ### Author Response · Authors · 2024-11-22
>
> We thank the reviewer for their detailed read of our submission. Below, we provide clarifications regarding the questions asked and the comments on our theoretical analysis.
>
> In addition to the below, we welcome the reviewer to elaborate on the originality concerns, such as past works we may have omitted. To the best of our knowledge, the SSL community has focused on dimension orthogonality, but the connection between NCE objectives and origin centering in our work is novel.
>
> ## Questions
>
> ### Question 1
> A: We agree that an important observation in our work is that for some real-world graphs applying only positive updates without any repulsion performs well. As the reviewer suggests, in the  case of $II^0$, a small number of epochs are used to prevent embedding collapse. The number of epochs is tuned using the validation edge set.
>
> ### Question 2
> A: Yes, the method can be extended to the weighted setting. For instance, one can sample positive edges such that the probability of selecting an edge is proportional to its weight. Repulsion would still be applied via dimension regularization as proposed in our augmentation.
>
> ## Comments
>
> > For instance, in the proof of Proposition 2.2, it is not clear how we obtain the equation given in Line 730 from Equation 27.
>
> We thank the reviewer for this comment and note it is due to a difficulty in the readability of our notation. The constant $C$ in line 730 ($<X_i^t, X_j^t> + C$) is different from the constriction symbol $\mathcal{C}$. We will make this distinction clearer in the Appendix.
>
> >  In Lines 731-732, the authors state that the constriction is monotonically increasing, since it remains positive. But this argument doesn’t guarantee that we will obtain C ≥ c for any given c.
>
> We thank the reviewer for pointing out this correction. We will update Proposition 2.2 to state “As the Euclidean norm of the initial embeddings and the learning rate approach zero, there exists a time step $t$ such that applying gradient descent on $P_{SG}$ monotonically increases the constriction”. The revised claim still supports the argument that without repulsion the embeddings approach collapse.
>
> > However, Equation 11 holds since the denominator term goes to infinity so it does not establish the convergence of the gradients of $N_{SG}$ and $N’_{SG}$.
>
> We note that Proposition 2.3 is not trivial since even though the denominator approaches infinity, it is not obvious the numerator remains small in comparison. The numerator difference $\nabla  (N_{SG}' - N_{SG})$ ) corresponds to the repulsion of positive pairs, which one would expect to be large in magnitude. In the proof we show that when the embedding norm is bounded (not infinitely large), the sparsity of the graph ensures that the repulsion between positive pairs is small relative to the repulsion among negative pairs.
>
> > The derivation from Equation 37 to Equation 35 is not clearly explained.
>
> In short, the transition from (35) to (36) is the triangle inequality. From (36) to (37), we utilize the following: $K_{ij} \leq \mathcal{C}$; $\| X_i\| \leq \beta_{max}$; the number of negative pairs is $n^2 - m$. We are happy to elaborate as needed.
>
> We thank the reviewer for their notes on notation and will incorporate the recommended changes.

---

> > ### Author Response · Authors · 2024-11-27
> >
> > As a follow up, we have uploaded a revised PDF that accounts for the comment on Proposition 2.2 as well as the recommended notation changes.

---

> > > ### Comment · Reviewer_Fdqs · 2024-12-03
> > > **Thank you for the updates**
> > >
> > > I thank the authors for responding to my questions and comments, and I raised my score, but I still keep some of my concerns related to the theoretical contributions and experimental evaluation. For instance, the numerator term in Proposition 2.3. is upper bounded by the term $m\beta_{max}$ where $\beta_{max} \geq 1$ is a constant and $m$ is the number of non-zero entries in the sparse (assumption) similarity matrix $S$. Therefore, having an upper bound on the numerator term doesn't guarantee that it will "remain small". $m$ can be a very big number so the difference $|| ∇N^{′} SG − ∇NS ||$ can still be large.

---

### Meta-Review · Area_Chair_GFhi · 2024-12-20

**Metareview:**

This work introduces a scalable framework that replaces SGNS with dimension regularization, improving efficiency while maintaining or enhancing performance in tasks like link prediction.

There is consensus among the reviewers that the paper is marginally below acceptance. I summarize some issues below.

1) Lack of novelty, the paper is a simple application of existing SSL techniques to a different setting. The connection between NCE loss and dimension regularization is new, but at the same time a trivial application.
2) Limited applicability due to performance drop reported by all reviewers.
2) Better experimental evaluation is needed, e.g., more comparisons with baselines. Also reported by all reviewers. In particular, Fdqs asked for comparison to VERSE. Kz8w asked for a comparison of SGNS to the full gradient solution (which presumably is better than SGNS. Reviewer ex3Z, asked for some experiment results to justify how this can be potentially useful for skip-gram based language models.
3) Better description of the theoretical analysis. Especially, Reviewer Fdqs raised an important concern regarding the usefulness of the provided bounds. The theoretical analysis needs to be potentially improved or provide further clarification.

**Additional Comments On Reviewer Discussion:**

See my comments above.

---

### Decision · Program_Chairs · 2025-01-22

Reject